# Learning to Zoom with Anatomical Relations for Medical Structure Detection

**Bin Pu[1], Liwen Wang[2], Xingbo Dong[2], Xingguo Lv[1], Zhe Jin[2]\***
[1]Hunan University [2]Anhui University
pubin@hnu.edu.cn, liwenwang@stu.ahu.edu.cn
xingbo.dong@ahu.edu.cn, lvxg@hnu.edu.cn, zhejin@ahu.edu.cn

## Abstract

Accurate anatomical structure detection is a critical preliminary step for diagnosing diseases characterized by structural abnormalities. In clinical practice, medical experts frequently adjust the zoom level of medical images to obtain comprehensive views for diagnosis. This common interaction results in significant variations in the apparent scale of anatomical structures across different images or fields of view. However, the information embedded in these zoom-induced scale changes is often overlooked by existing detection algorithms. In addition, human organs possess a priori, fixed topological knowledge. To overcome this limitation, we propose ZR-DETR, a zoom-aware probabilistic framework tailored for medical object detection. ZR-DETR uniquely incorporates scale-sensitive zoom embeddings, anatomical relation constraints, and a Gaussian Process-based detection head. This architecture enables the framework to jointly model semantic context, enforce anatomical plausibility, and quantify detection uncertainty. Empirical validation across three diverse medical imaging benchmarks demonstrates that ZR-DETR consistently outperforms strong baselines in both single-domain and unsupervised domain adaptation scenarios.

## 1 Introduction

Anatomical structure detection in medical imaging is a fundamental task in disease diagnosis, playing a pivotal role in identifying anatomical abnormalities and pathological conditions [1]. In clinical practice, radiologists routinely adjust zoom levels and viewing angles when interpreting images acquired from modalities such as CT, MRI, or ultrasound [2, 3]. These adjustments, inherently subjective and experience-driven, lead to considerable variability in image scale and perspective across different examinations [2]. As a result, object detection models often encounter difficulties in generalizing across varying organ sizes and morphological presentations [4].

To address the challenge of scale variation, previous work has explored explicit zoom estimation and normalization strategies [2]. For instance, certain methods predict the zoom ratio of the input image and manually rescale it to ensure a uniform organ scale across samples. Parallel efforts, such as Feature Pyramid Networks (FPN) [5], Deformable-DETR [6] and Dynamic Zoom-in Network [7], focus on multi-scale feature extraction to improve detection robustness across different object sizes. However, these approaches typically neglect the structural priors inherent in anatomical configurations, which are crucial for reliable localization in clinical imaging scenarios.

Incorporation of structural relation priors has been attempted through graph-based morphological modeling and alignment techniques [8, 9, 10, 11]. While graph matching can facilitate structural correspondence across images, it is highly sensitive to noise and dependent on image quality, thereby

---

\*Corresponding Author

39th Conference on Neural Information Processing Systems (NeurIPS 2025).

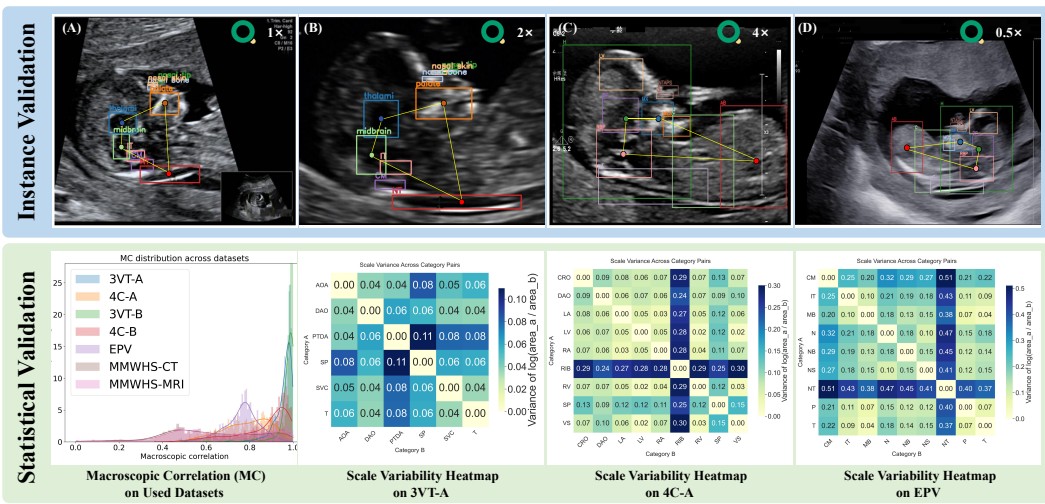

Figure 1: Motivation of our porposed ZR-DETR. The upper row illustrates the observed structural relation consistency and zoom patterns in our proposed ZR-DETR framework. The second row provides empirical validation through visualizations of Macroscopic Correlation (MC) across the employed datasets and category-wise scale variability heatmaps.

limiting its effectiveness under real-world clinical variability. Consequently, such methods remain vulnerable to artifacts introduced by inconsistent zoom levels, resolution, and imaging noise.

We analyze structural consistency and scale variability across several medical imaging datasets, as shown in Fig. 1. To quantify the spatial regularity between anatomical targets, we compute the *Macroscopic Correlation* (MC) and find that most datasets exhibit high MC values close to 1.0, indicating strong geometric consistency in organ placement. Furthermore, we measure the variance of normalized area ratios across object categories and observe that the majority of values remain below 1.0 across all datasets. This suggests that, although absolute object sizes may vary due to factors such as patient anatomy or imaging protocol, their relative scales are largely preserved within each domain. These empirical findings confirm the stability of anatomical structural relationships and reveal distinct scale distribution patterns among object categories, thereby reinforcing the motivation for our zoom-aware, anatomy-constrained detection framework.

To address the challenges of morphological variability, structural inconsistency, and uncertainty in medical object detection, we propose a unified framework named ZR-DETR, which integrates deterministic modeling with probabilistic inference. ZR-DETR exhibits several key innovations: **1)** ZR-DETR introduces a Zoom Relation Encoder that captures relational zoom patterns among object proposals in the latent space, enabling the network to learn scale-aware structural priors and adapt to variability in organ sizes across different imaging conditions; **2)** ZR-DETR incorporates anatomical relation consistency constraints into the training objective, which encode spatial dependencies among organ classes and effectively preserve plausible anatomical topology by penalizing geometrically implausible predictions; **3)** ZR-DETR employs a Gaussian Process-based detection head, which models both the predictive mean and uncertainty by fusing visual, scale, and anatomical priors into a unified kernel space, enabling principled uncertainty quantification and improved calibration under limited-data and cross-domain scenarios. **Our key contributions are summarized as follows:**

**1.** In accordance with the identified Zoom Pattern, we developed the Zoom Relation Encoder to effectively represent the organ zooming behavior in medical imaging. This encoder is intended to direct the model's attention towards the relevant features and dimensions of the specific organ that is to be identified. Concurrently, we implemented Anatomical Relation Consistency Constraints to ensure that the structural similarity between the detection outcomes and the actual annotations is maintained, drawing upon the structural priors that have been acquired through the learning process.

**2.** We propose a probabilistic detection framework utilizing Gaussian processes, which imposes constraints on prediction outcomes across three dimensions: appearance features, zoom level, and structural consistency. This is achieved through the design of specialized kernel functions, which also facilitate the assessment of confidence and uncertainty associated with the detection results.

**3.** Extensive experiments across diverse medical imaging benchmarks demonstrate that our approach consistently surpasses robust baseline models in terms of detection accuracy and uncertainty calibration, especially within the context of unsupervised domain adaptation (UDA) scenarios.

## 2 Related Works

### 2.1 Zoom Pattern in Medical Images

Magnifying images to enhance anatomical visualization is essential across medical modalities. This is particularly critical in diagnostic procedures like fetal ultrasound (for CRL/NT measurements) and broader disciplines (radiology, cardiology, oncology), where dynamic zooming helps resolve regions of interest, identify pathologies, and ensure accuracy, whether assessing tumor margins in MRI, vascular structures in CT angiography, or cardiac motion in echocardiography [2, 3, 4].

Several previous approaches have attempted to utilize this information to facilitate the analysis of medical images [12, 13]. For instance, [14] employs magnification as a key technique in the classification of histopathological images through their multi-scale approach. Others attempt to identify the optimal magnification level for histopathological images, in order to determine the magnification level at which the best performance can be obtained when training convolutional neural networks to detect breast cancer in histopathological images [15, 16]. There are other ways to consider the zoom level of an image. For example, predicting whether or not the entire chest silhouette area is visible within the US fan-shaped area of the image [17]. [18] introduces a multi-scale strategy that combines multi-scale feature extraction with a scale-aware test-time adaptation mechanism, enabling the model to dynamically adjust its receptive field based on lesion size and thereby handle the variability and scale diversity of small to large lesions. [2] explores how to use this zoom information, which is an under-utilised piece of information that is extractable from fetal ultrasound images, and explores associating zooming patterns with specific structures to improve the structure detection.

### 2.2 Relation Modeling for Object Detection

Rather than modeling visual relations at pixel, patch, or image levels, relation networks capture interactions at the instance level, enabling finer-grained relational reasoning. Existing studies on relation modeling can be broadly categorized into category-based and instance-based approaches. Category-based methods construct conceptual or statistical relations, such as co-occurrence probabilities [19, 20], either from external datasets like Visual Genome [19, 21, 22] or by learning from class labels in a data-driven manner [20]. However, these methods introduce additional complexity due to the necessity of instance-to-category assignments [22, 20, 23, 24].

In contrast, instance-based approaches directly model object-level relations by constructing a graph where each object proposal is a node and pairwise relations form the edges. Such graph structures allow relational reasoning to be integrated into the training process, with relation weights learned explicitly [25, 26]. These weights typically measure high-dimensional distances between object pairs, such as appearance similarity [27], spatial proximity [28], or attention-based affinity [29, 11]. Nonetheless, relying solely on self-attention weights learned from data without structural priors increases the demand for large-scale datasets and prolonged training. To alleviate this, we introduce zoom patterns as inductive priors, aiming to reduce data requirements and improve learning efficiency.

## 3 Methodology

### 3.1 Framework Overview

As depicted in Fig. 2, the proposed architecture follows a two-branch detection pipeline inspired by Deformable DETR [6], consisting of a Transformer-based encoder and a probabilistic Gaussian Process (GP)-based detection head. Given an input image $\mathcal{I} \in \mathbb{R}^{H \times W \times 3}$, a ResNet-50 backbone extracts multi-scale features $\{\mathbf{F}_l\}_{l=1}^4$, which are flattened and projected into a token matrix $\mathbf{Z} \in \mathbb{R}^{T \times D}$. These tokens are processed by $L$ layers of multi-head self-attention:

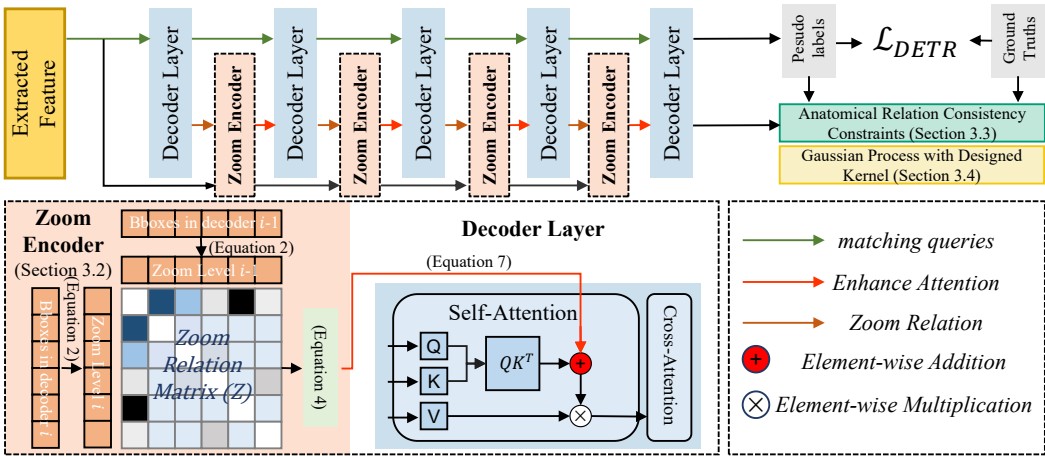

Figure 2: ZR-DETR integrates a Transformer-based encoder-decoder architecture, incorporating Zoom Embeddings to process multi-scale medical image features, Anatomical Relation Consistency Constraints to encode prior anatomical knowledge, and a Gaussian Process with a designed kernel for uncertainty-aware detection.

$$\mathbf{Z}^{(l)} = \text{LayerNorm}\Big(\mathbf{A}^{(l)} + \text{FFN}(\mathbf{A}^{(l)})\Big), \quad \mathbf{A}^{(l)} = \text{Softmax}\left(\frac{\mathbf{Q}^{(l)}\mathbf{K}^{(l)\top}}{\sqrt{D}}\right)\mathbf{V}^{(l)}, \qquad (1)$$

where $\mathbf{Q}^{(l)}$, $\mathbf{K}^{(l)}$, and $\mathbf{V}^{(l)}$ denote the query, key, and value matrices at the $l$-th layer, respectively; $D$ is the feature dimension; $\text{FFN}(\cdot)$ represents the feed-forward network; and $\text{LayerNorm}(\cdot)$ denotes layer normalization.

To enhance anatomical awareness, we augment $\mathbf{Z}$ with scale-sensitive embeddings that encode zoom-level patterns (see Section 3.2). For detection, the encoded tokens are decoded into object queries, and predictions are generated by a Gaussian Process (GP) head. Unlike deterministic detection heads, the GP outputs both a predictive mean and variance, enabling uncertainty-aware localization. The GP kernel fuses three complementary priors: (i) an RBF kernel over visual appearance features, (ii) a rational quadratic kernel over zoom embeddings, and (iii) a delta (Kronecker) kernel over anatomical class labels. This design enforces multi-scale structural consistency while preserving probabilistic interpretability. Matching supervision follows Relation-DETR [30], aligning predicted queries with ground-truth objects via Hungarian bipartite matching [31].

## 3.2 Zoom Relation Encoder

Inspired by the position relation encoder in Relation-DETR [30], we propose a *Zoom Relation Encoder* to introduce scale-aware structural priors into the attention mechanism. Rather than modeling pairwise spatial geometry, we construct relative relations between proposals based on their *log-scale zoom levels*, which characterize anatomical scale hierarchies more explicitly.

**Zoom Relation Encoding.** To capture the relative scale relations between object proposals, we define a zoom-level embedding that reflects the area-based contextual hierarchy among regions. Given a region proposal with area $A_i$, we define its zoom level as the logarithmic ratio between the proposal area and the image area:

$$\lambda_i = \log\left(\frac{A_i}{A_{\text{img}}}\right), \quad \lambda_i \in \left[\log\left(\frac{1}{W \cdot H}\right), \log(1)\right], \qquad (2)$$

where $A_{\text{img}}$ denotes the total image area. We then define the pairwise zoom-level relation between two proposals $i$ and $j$ as the ratio of their absolute zoom levels:

$$z_{ij} = \frac{|\lambda_i|}{|\lambda_j|}, \qquad (3)$$

which captures the relative scale of proposal $i$ with respect to $j$. A value of $z_{ij} > 1$ indicates that region $i$ is relatively larger than region $j$ in log scale, while $z_{ij} < 1$ indicates the opposite. To

transform this scalar relation into a high-dimensional vector suitable for attention modulation, we apply a sinusoidal positional encoding inspired by the original Transformer formulation:

$$\text{Embed}(z_{ij}, 2k) = \sin\left(\frac{s \cdot z_{ij}}{T^{2k/d_z}}\right), \tag{4}$$

$$\text{Embed}(z_{ij}, 2k+1) = \cos\left(\frac{s \cdot z_{ij}}{T^{2k/d_z}}\right),$$

where $s$ is a frequency scaling factor, $T$ is a temperature constant (typically $T = 10^4$), $d_z$ is the embedding dimensionality, and $k$ ranges from 0 to $d_z/2 - 1$. This yields a zoom-aware encoding tensor $E \in \mathbb{R}^{N \times N \times d_z}$ for $N$ proposals. We then apply a learnable linear projection to map $E$ to match the number of attention heads $M$:

$$\text{Zoom}(b_1, b_2) = \max(\epsilon, WE + B) \in \mathbb{R}^{N \times N \times M}, \tag{5}$$

where $W \in \mathbb{R}^{d_z \times M}$ and $B \in \mathbb{R}^M$ are learnable parameters, and $\epsilon$ is a small positive constant used to prevent degenerate values. The resulting tensor $\text{Zoom}(b_1, b_2)$ serves as a multiplicative modulation factor to enhance multi-head self-attention based on relative scale cues for proposal indices $b_1$ and $b_2$.

**Zoom Relation Decoder.** Following Deformable-DETR [6], we adopt a dual-branch decoder: one for deduplicated matching (*matching queries $Q_m$*) and one for rich positive supervision (*hybrid queries $Q_h$*). The self-attention mechanisms for the two branches are computed as:

$$\text{Attn}(Q_m^l) = \text{Softmax}\left(\frac{Q_m Q_m^\top}{\sqrt{d_{\text{model}}}} + \text{Zoom}(b^{l-1}, b^l)\right) Q_m, \tag{6}$$

$$\text{Attn}(Q_h^l) = \text{Softmax}\left(\frac{Q_h Q_h^\top}{\sqrt{d_{\text{model}}}}\right) Q_h.$$

This contrastive architecture enables the model to exploit zoom-level cues for scale-aware deduplication while maintaining diverse learning signals for enhanced convergence.

## 3.3 Anatomical Relation Consistency Constraints

Recognizing the highly structured nature of human anatomy, we explicitly embed prior spatial relationships among organs into the model. Specifically, we define an anatomical adjacency matrix $\mathbf{M} \in \{0, 1\}^{|\mathcal{A}| \times |\mathcal{A}|}$, where $\mathcal{A}$ denotes the set of anatomical classes. The entry $\mathbf{M}_{a_i, a_j} = 1$ indicates that organs $a_i$ and $a_j$ are anatomically adjacent or frequently co-occur, based on population-level statistics or anatomical atlases.

To enforce this prior during training, we introduce an anatomical relation consistency loss that encourages geometric plausibility between predicted organ locations:

$$\mathcal{L}_{\text{anatomy}} = \frac{1}{|\mathcal{P}|} \sum_{i=1}^{T} \sum_{j=1}^{T} \mathbf{M}_{a_i, a_j} \left\| k_{\text{spatial}}(\mathbf{b}_i, \mathbf{b}_j) - k_{\text{prior}}(a_i, a_j) \right\|_2, \tag{7}$$

where $\mathbf{b}_i, \mathbf{b}_j$ are the predicted bounding boxes for instances $i$ and $j$, $a_i, a_j$ their corresponding anatomical classes, and $\mathcal{P} = \{(i, j) \mid \mathbf{M}_{a_i, a_j} = 1\}$ is the set of valid (i.e., anatomically related) organ pairs. The target value $k_{\text{prior}}(a_i, a_j) \in [0, 1]$ represents the expected normalized spatial proximity between organs $a_i$ and $a_j$, estimated from the training data.

The spatial kernel $k_{\text{spatial}}$ quantifies the observed proximity between two bounding boxes using a normalized Hausdorff distance:

$$k_{\text{spatial}}(\mathbf{b}_i, \mathbf{b}_j) = 1 - \frac{d_H(\mathbf{b}_i, \mathbf{b}_j)}{\text{diag}(\mathbf{b}_i \cup \mathbf{b}_j)}, \tag{8}$$

where $\text{diag}(\mathbf{b}_i \cup \mathbf{b}_j)$ denotes the length of the diagonal of the smallest enclosing bounding box of $\mathbf{b}_i$ and $\mathbf{b}_j$, and the Hausdorff distance $d_H$ is defined as:

$$d_H(\mathbf{b}_i, \mathbf{b}_j) = \max\left\{ \sup_{p \in \partial \mathbf{b}_i} \inf_{q \in \partial \mathbf{b}_j} \|p - q\|_2, \ \sup_{q \in \partial \mathbf{b}_j} \inf_{p \in \partial \mathbf{b}_i} \|p - q\|_2 \right\}, \tag{9}$$

with $\partial\mathbf{b}$ denoting the boundary of bounding box $\mathbf{b}$. This formulation ensures that predictions violating known anatomical spatial constraints are penalized, thereby improving detection consistency and anatomical plausibility.

**Proof of Positive Definiteness:** For finite point sets $\mathcal{P}, \mathcal{Q}$, the Hausdorff metric satisfies:

1. $d_H(\mathcal{P}, \mathcal{Q}) \geq 0$,
2. $d_H(\mathcal{P}, \mathcal{Q}) = 0 \iff \mathcal{P} = \mathcal{Q}$,
3. Triangle inequality: $d_H(\mathcal{P}, \mathcal{R}) \leq d_H(\mathcal{P}, \mathcal{Q}) + d_H(\mathcal{Q}, \mathcal{R})$.

Thus, $k_{\text{spatial}} = \exp(-\gamma d_H^2)$ is a valid positive definite kernel by Schoenberg's theorem.

## 3.4 Probabilistic Detection with Gaussian Processes

To quantify predictive uncertainty and incorporate nonparametric function modeling, we introduce a Gaussian Process (GP)-based detection head. In this approach, each detection score is modeled as a sample from a stochastic function $f \sim \mathcal{GP}(0, \mathbf{K})$, where $\mathbf{K}$ is the kernel matrix defined over predicted and ground-truth bounding boxes. The kernel function integrates visual appearance, scale, and anatomical identity to define a structured similarity space for detection outputs. Specifically, we construct $\mathbf{K}$ using the ground-truth bounding boxes $\{b_i^*\}$, which capture the intrinsic structural characteristics of the target anatomy.

The composite kernel is defined as the sum of three interpretable components:

$$k(b_i, b_j) = \underbrace{\sigma_f^2 \exp\left(-\frac{\|f_i - f_j\|^2}{2\ell_c^2}\right)}_{\text{(a) appearance}} + \underbrace{\sigma_z^2\left(1 + \frac{(|\lambda_i| - |\lambda_j|)^2}{2\alpha\ell_s^2}\right)^{-\alpha}}_{\text{(b) scale (zoom level)}} + \underbrace{\sigma_a^2 \mathbb{I}(a_i = a_j)}_{\text{(c) anatomy}}, \qquad (10)$$

where $\sigma_f^2$, $\sigma_z^2$, and $\sigma_a^2$ are learnable weights that balance the contributions of appearance, scale, and anatomical priors. Here, $f_i$ denotes the visual feature vector extracted from the image region corresponding to bounding box $b_i$, and $\lambda_i = \log(A_i/A_{\text{img}})$ is its log-scale zoom level as defined in Eq. (2).

The first term is a Radial Basis Function (RBF) kernel operating on appearance features with length scale $\ell_c$. The second term is a rational quadratic kernel that provides robustness to variations in object scale by modeling the zoom-level difference with heavy-tailed sensitivity controlled by $\alpha$. The third term is a discrete Kronecker delta kernel ($\mathbb{I}(\cdot)$ is the indicator function) that enforces anatomical consistency—assigning non-zero similarity only between proposals of the same anatomical class. This is particularly critical in medical imaging, where anatomical semantics strongly constrain plausible detections.

For a test input $\tilde{\mathbf{z}}_*$, the predicted class probability is obtained by marginalizing over the GP posterior and applying the softmax function:

$$p(y_* = c \mid \tilde{\mathbf{z}}_*, \mathcal{D}) = \frac{\exp(\mu_*^{(c)})}{\sum_{c'} \exp(\mu_*^{(c')})}, \qquad (11)$$

where $\mu_*^{(c)}$ is the posterior mean for class $c$. The posterior mean and variance are computed as:

$$\mu_*^{(c)} = \mathbf{k}_*^{(c)\top}(\mathbf{K} + \sigma_n^2\mathbf{I})^{-1}\mathbf{y}, \quad \sigma_*^{2(c)} = k_{**}^{(c)} - \mathbf{k}_*^{(c)\top}(\mathbf{K} + \sigma_n^2\mathbf{I})^{-1}\mathbf{k}_*^{(c)}. \qquad (12)$$

Here, $\mathbf{K}$ is the kernel matrix evaluated on the training set (using ground-truth boxes), $\mathbf{k}_*^{(c)}$ is the kernel vector between the test input and all training samples of class $c$, $k_{**}^{(c)} = k(\tilde{\mathbf{z}}_*, \tilde{\mathbf{z}}_*)$ is the self-kernel value, $\mathbf{y}$ is the one-hot label vector, and $\sigma_n^2$ models observation noise. This formulation yields well-calibrated probabilistic predictions that reflect both epistemic uncertainty and structural anatomical priors.

## 3.5 Integrated Optimization Strategy

The overall training objective combines multiple loss components as follows:

$$\mathcal{L}_{\text{supervised}} = \mathcal{L}_{\text{DETR}} + \underbrace{\beta_1 \mathcal{L}_{\text{anatomy}}}_{\text{anatomy consistency}} + \underbrace{\beta_2 \, \text{KL}(\mathcal{N}(\mathbf{m}, \mathbf{S}) \,\|\, \mathcal{N}(0, \mathbf{K}_{\text{GT}}))}_{\text{GP consistency}}, \qquad (13)$$

where $\mathcal{N}(\cdot, \cdot)$ denotes a Gaussian distribution. The first term corresponds to the standard detection loss in DETR [6]. Following Relation-DETR [30], we adopt a dual-branch supervised loss:

$$\mathcal{L}_{\text{DETR}} = \mathcal{L}_m(\boldsymbol{p}_m, \boldsymbol{g}) + \mathcal{L}_h(\boldsymbol{p}_h, \boldsymbol{g}), \tag{14}$$

where $\boldsymbol{p}_m$ and $\boldsymbol{p}_h$ denote the predictions from the matching and hybrid branches, respectively, and $\boldsymbol{g}$ represents the ground-truth targets. Both $\mathcal{L}_m$ and $\mathcal{L}_h$ follow the formulation of H-DETR [32].

The second term, $\mathcal{L}_{\text{anatomy}}$, enforces anatomical consistency across detected instances, which is particularly beneficial under domain shifts. The third term introduces a novel GP-based regularization that aligns the posterior distribution of predicted bounding boxes with the ground-truth structure in kernel space. This posterior is parameterized via variational inference:

$$\mathbf{m} = \arg\min_{\mathbf{m}} \|\mathbf{y} - \sigma(\mathbf{m})\|_2^2 + \text{tr}(\mathbf{S}), \quad \mathbf{S} = \text{diag}\left(\sigma_n^2 \left(\mathbf{K}^{-1} + \epsilon\mathbf{I}\right)^{-1}\right), \tag{15}$$

where $\epsilon > 0$ ensures numerical stability, $\sigma(\cdot)$ is the sigmoid function, and $\mathbf{y}$ is the one-hot label vector. The hyperparameters $\beta_1 = 0.5$ and $\beta_2 = 0.1$ are selected via grid search on a validation set.

In the unsupervised domain adaptation (UDA) setting, we retain the same network architecture as in the single-domain experiment. During training on target-domain samples, we further impose consistency constraints using the Gaussian Process kernel matrix defined in Equation (10). The UDA objective is formulated as:

$$\mathcal{L}_{\text{UDA}} = \mathcal{L}_{\text{supervised}} + \text{KL}(\mathcal{N}(\mathbf{m}_t, \mathbf{S}_t) \,\|\, \mathcal{N}(0, \mathbf{K}_{\text{GT}})), \tag{16}$$

where $\mathbf{m}_t$ and $\mathbf{S}_t$ are the variational parameters (mean and covariance) computed from target-domain predictions, and $\mathcal{N}(\cdot, \cdot)$ again denotes a Gaussian distribution.

## 4 Experiments

### 4.1 Dataset

**Fetal Cardiac Structure (FCS) [9]** is a diversified ultrasound dataset collected from two medical centers, each containing two views of the heart, *i.e.*, three vessels and trachea view (3VT) and four-chamber cardiac view (4C). These datasets are from different medical devices, such as Samsung, Sonoscape, and Philip, with a gestational week range of 20-34 weeks. The 3VT and 4C from *A* medical center are denoted as **3VT-A** and **4C-A**. Similarly, they from *B* medical center are denoted as **3VT-B** and **4C-B**. The total number of 4C-A, 4C-B, 3VT-A, and 3VT-B are 810, 809, 891, and 369, respectively. 4C contains 9 anatomical structures, *i.e.*, Left ventricle (LV), Left atrium (LA), Right ventricle (RV), Descending aorta (DAO), Right atrium (RA), Ventricular septum (VS), Spine (SP), Rib (RIB), and Cross (CRO). 3VT contains Superior vena cava (SVC), Arch of Aorta (AOA), Trachea (T), SP, Pulmonary trunk & ductus arteriosus (PTDA), and DAO.

**Early Pregnancy View (EPV) [33]** is a challenging early pregnancy ultrasound dataset collected from different ultrasound devices, a total number of 1131 images, and its gestational range is 10-14 weeks. EP includes 9 key structures, *i.e.*, thalami (TH), midbrain (MB), palate (PAL), Intracranial Transparent (IT, *i.e.*, 4th ventricle), cisterna magna (CM), nuchal translucency (NT), nasal tip (NST), nasal skin (NS), and nasal bone (NB).

**MM-WHS [34]** consists of 20 unpaired MRI and 20 CT volumes with corresponding pixel-level segmentation ground truth. We use the pre-processing methods of PnP-AdaNet [35] and convert the segmentation masks into bounding boxes for four regions present in both MRI and CT modalities: the ascending aorta (AA), the left atrial blood cavity (LA-blood), the left ventricular blood cavity (LV-blood) and the left ventricular myocardium (LV-MYO).

### 4.2 Implementation Details

For a fair comparison, we use ResNet-50 [36] as the backbone for all experiments, which is implemented in PyTorch and trained for 20 epochs and 2 batch size with one RTX3090 GPU. For data augmentation, we use random horizontal flipping, random color jittering, grayscale, gaussian blurring, and cutout patches for image augmentation. We uniformly resized medical images to 800×1333 for all stages, and we trained the model using the AdamW optimizer with an initial learning rate of 0.01 with the weight decay of $1\times10^{-4}$. The FCS, EPV and MM-WHS datasets were divided into a

training set, a validation set, and a test set in the ratio of 7:1:2, and all the settings remained the same. During the evaluation phase, we report the Average Precision (AP) across all test datasets using an Intersection over Union (IoU) threshold of 0.5. Results are presented in terms of $AP_{50}$ (%) and the overall mean average precision (mAP).

### 4.3 Experimental Results

In content of single domain structure detection tesk, We compare our ZR-DETR with seven competing object detection methods. These include a tranditional method FasterRCNN [37] and six high-related methods based on transformer (Deformable-DETR [6], DAB-DETR [38], DN-DETR [39], Relation-DETR [30], MI-DETR [40]).

**Comparison on the FCS (4C) dataset.** As shown in Table 1, on Site A, ZR-DETR achieves a leading 97.3% mAP, with particularly strong performance on anatomically complex structures DAO: 98.7%, PTDA: 98.7%. This reflects the effectiveness of our anatomical prior integration and zoom-aware attention mechanism. The model maintains robust performance on Site B 48.8% mAP, showing significant advantages in challenging cases SVC: 73.2%, AOA: 61.2%, while other methods exhibit severe performance degradation (*e.g.*, DINO drops to 5.18% on DAO detection). In the challenging A → B

Table 1: The performance of different detection methods in FCS (4C) dataset [9].

| Method | LA ↑ | RA ↑ | LV ↑ | RV ↑ | VS ↑ | CRO ↑ | SP ↑ | DAO ↑ | RIB ↑ | mAP ↑ |
|---|---|---|---|---|---|---|---|---|---|---|
| Single Domain Structure Detection (Site A) | | | | | | | | | | |
| FasterRCNN [37] CVPR16 | 92.0 | 96.6 | 94.1 | 91.5 | 96.9 | 97.0 | 94.2 | 93.1 | 73.0 | 92.1 |
| Deformable-DETR [6] ICLR21 | 94.5 | 95.6 | 96.0 | 95.2 | 96.7 | 93.1 | 97.5 | 97.4 | 73.5 | 93.3 |
| DAB-DETR [38] ICLR22 | 96.5 | 96.7 | 95.6 | 94.5 | 97.5 | 98.5 | 97.5 | 98.0 | 76.8 | 94.6 |
| DN-DETR [39] CVPR22 | 97.2 | 98.0 | 98.9 | 97.7 | 99.8 | 97.7 | 99.2 | 98.5 | 85.4 | 96.9 |
| DINO [41] ICLR23 | 97.3 | 97.8 | 99.4 | 97.5 | 99.9 | 97.9 | 98.1 | 98.5 | 86.4 | 97.0 |
| Relation-DETR [30] ECCV24 | 97.6 | 98.4 | 97.8 | 97.2 | 99.6 | 97.9 | 99.9 | 99.6 | 87.8 | 97.3 |
| ZR-DETR (Ours) | 98.1 | 97.9 | 98.9 | 99.6 | 99.9 | 100 | 99.9 | 99.6 | 87.7 | 98.0 |
| Single Domain Structure Detection (Site B) | | | | | | | | | | |
| FasterRCNN [37] CVPR16 | 71.3 | 91.3 | 86.7 | 80.5 | 90.4 | 87.4 | 88.1 | 89.1 | 82.6 | 85.3 |
| Deformable-DETR [6] ICLR21 | 75.7 | 87.4 | 90.7 | 85.2 | 91.0 | 85.0 | 90.0 | 91.0 | 84.3 | 86.7 |
| DAB-DETR [38] ICLR22 | 75.5 | 92.2 | 92.9 | 88.4 | 94.3 | 86.7 | 92.1 | 92.1 | 86.6 | 89.0 |
| DN-DETR [39] CVPR22 | 77.1 | 92.3 | 92.0 | 87.6 | 96.0 | 87.9 | 93.2 | 92.4 | 85.0 | 89.3 |
| DINO [41] ICLR23 | 83.4 | 95.2 | 96.4 | 92.9 | 97.0 | 92.2 | 94.3 | 93.0 | 89.5 | 92.7 |
| Relation-DETR [30] ECCV24 | 87.9 | 95.1 | 97.5 | 94.2 | 94.9 | 92.7 | 96.6 | 93.3 | 88.4 | 93.4 |
| ZR-DETR (Ours) | 89.0 | 95.0 | 97.4 | 95.5 | 96.1 | 94.0 | 96.4 | 94.4 | 90.0 | 94.2 |
| Cross Domain Structure Detection (A→B) | | | | | | | | | | |
| SIGMA [42] CVPR22 | 50.1 | 62.1 | 49.5 | 51.3 | 58.9 | 55.6 | 46.7 | 54.0 | 47.9 | 52.0 |
| SIGMA++ [43] TPAMI 23 | 57.1 | 57.0 | 60.2 | 58.3 | 56.1 | 58.9 | 55.5 | 59.1 | 60.1 | 57.8 |
| $M^3$-UDA [9] CVPR24 | 79.9 | 69.8 | 72.8 | 71.7 | 81.0 | 78.0 | 81.7 | 78.0 | 78.3 | 76.8 |
| ToMo-UDA [10] ICML24 | 64.2 | 75.6 | 70.4 | 64.3 | 66.7 | 75.0 | 75.5 | 77.2 | 73.0 | 71.3 |
| DATR [44] TIP25 | 81.4 | 62.4 | 68.9 | 74.1 | 81.8 | 73.9 | 82.6 | 83.7 | 62.3 | 74.6 |
| ZR-DETR-UDA (Ours) | 58.4 | 82.2 | 80.8 | 74.6 | 89.5 | 83.2 | 83.4 | 87.4 | 78.2 | 79.8 |

UDA scenario, ZR-DETR-UDA establishes new state-of-the-art performance 79.8% mAP, outperforming strong baselines like $M^3$-UDA 76.8%. The improvements are particularly notable in SP, PTDA, and T scores, demonstrating the effectiveness of our probabilistic modeling approach and multi-scale adaptation strategy.

**Comparison on the FCS (3VT) dataset.** As shown in Table 2 and 6, we evaluate ZR-DETR and its UDA variant under three scenarios: Single-Domain Detection (Site A and Site B) and Cross-Domain Detection (A→B). ZR-DETR demonstrates consistently superior performance across all evaluated settings. On Site A, it achieves the highest mAP 97.3% with notable improvements in DAO 98.7% and PTDA 98.7%, reflecting precise structural modeling enabled by anatomical priors and zoom-aware attention. Despite its accuracy, ZR-DETR remains computationally efficient (45.172ms, 115.4M, 0.268 TFLOPs), outperforming heavier models such as MI-DETR. On Site B, where domain shift increases task difficulty, ZR-DETR still leads mAP 48.8%, showing robustness with significant gains in SVC 73.2%

Table 2: The performance of different detection methods in FCS (3VT) dataset [9].

| Method | DAO ↑ | SP ↑ | PTDA ↑ | T ↑ | SVC ↑ | AOA ↑ | mAP ↑ |
|---|---|---|---|---|---|---|---|
| Single Domain Structure Detection (Site A) | | | | | | | |
| FasterRCNN [37] CVPR16 | 93.7 | 92.0 | 96.9 | 87.1 | 87.5 | 94.5 | 91.9 |
| Deformable-DETR [6] ICLR21 | 97.1 | 90.2 | 96.9 | 91.4 | 91.0 | 98.8 | 94.2 |
| DAB-DETR [38] ICLR22 | 96.5 | 96.3 | 96.8 | 92.0 | 89.0 | 97.9 | 94.7 |
| DN-DETR [39] CVPR22 | 97.2 | 92.7 | 96.7 | 89.5 | 93.7 | 98.8 | 94.8 |
| DINO [41] ICLR23 | 95.8 | 97.4 | 97.7 | 94.5 | 93.0 | 99.4 | 96.6 |
| Relation-DETR [30] ECCV24 | 96.5 | 94.0 | 96.5 | 94.8 | 93.0 | 99.7 | 95.8 |
| MI-DETR [30] CVPR25 | 97.8 | 94.7 | 97.7 | 94.5 | 93.0 | 99.4 | 96.6 |
| ZR-DETR (Ours) | 98.7 | 96.9 | 98.7 | 95.3 | 94.3 | 99.8 | 97.3 |
| Single Domain Structure Detection (Site B) | | | | | | | |
| FasterRCNN [37] CVPR16 | 54.8 | 30.2 | 30.0 | 19.9 | 39.5 | 41.3 | 35.9 |
| Deformable-DETR [6] ICLR21 | 51.3 | 22.4 | 31.9 | 20.5 | 36.9 | 56.0 | 36.8 |
| DAB-DETR [38] ICLR22 | 44.6 | 35.7 | 29.6 | 27.2 | 37.8 | 62.4 | 39.6 |
| DN-DETR [39] CVPR22 | 51.9 | 33.4 | 31.0 | 23.4 | 44.9 | 60.0 | 40.8 |
| DINO [41] ICLR23 | 51.8 | 39.0 | 29.0 | 26.6 | 52.7 | 62.5 | 43.6 |
| Relation-DETR [30] ECCV24 | 56.9 | 42.2 | 32.6 | 22.5 | 57.9 | 56.1 | 44.7 |
| MI-DETR [30] CVPR25 | 53.2 | 43.2 | 32.7 | 22.3 | 62.4 | 54.8 | 44.8 |
| ZR-DETR (Ours) | 57.4 | 39.1 | 33.2 | 28.7 | 73.2 | 61.2 | 48.8 |
| Cross Domain Structure Detection (A→B) | | | | | | | |
| SIGMA [42] CVPR22 | 42.9 | 42.8 | 59.4 | 39.6 | 41.7 | 60.0 | 47.7 |
| SIGMA++ [43] TPAMI 23 | 42.3 | 37.4 | 45.4 | 29.0 | 32.0 | 42.9 | 38.1 |
| $M^3$-UDA [9] CVPR24 | 59.5 | 59.7 | 70.1 | 51.9 | 52.4 | 68.9 | 60.4 |
| ToMo-UDA [10] ICML24 | 45.4 | 60.1 | 81.5 | 27.6 | 45.6 | 63.7 | 54.0 |
| DATR [44] TIP25 | 57.4 | 59.2 | 61.7 | 56.9 | 50.7 | 62.7 | 58.1 |
| ZR-DETR-UDA (Ours) | 61.1 | 67.1 | 67.8 | 60.8 | 49.4 | 67.9 | 62.4 |

culty, ZR-DETR still leads mAP 48.8%, showing robustness with significant gains in SVC 73.2%

Table 4: The performance of different detection methods in MM-WHS dataset [34].

| Method | CT | | | | mAP ↑ | MRI | | | | mAP ↑ |
|---|---|---|---|---|---|---|---|---|---|---|
| | LV-MYO ↑ | LA-blood ↑ | LV-blood ↑ | AA ↑ | | LV-MYO ↑ | LA-blood ↑ | LV-blood ↑ | AA ↑ | |
| | | | | | **Single Domain Structure Detection** | | | | | |
| FasterRCNN [37] | 92.7 | 80.7 | 88.1 | 85.9 | 86.9 | 80.9 | 78.2 | 74.2 | 60.1 | 73.4 |
| Deformable-DETR [6] | 90.6 | 82.9 | 90.4 | 88.0 | 88.0 | 72.8 | 76.7 | 75.5 | 66.5 | 72.9 |
| DAB-DETR [38] | 91.3 | 83.2 | 86.7 | 93.8 | 88.8 | 87.1 | 80.5 | 74.9 | 57.1 | 74.9 |
| DN-DETR [39] | **96.3** | 86.5 | 89.9 | 89.0 | 89.4 | 84.7 | **85.8** | 75.4 | 66.1 | 78.0 |
| DINO [41] | 83.7 | 81.9 | 81.6 | 82.2 | 82.4 | 86.7 | 82.8 | 82.9 | 71.9 | 81.1 |
| Relation-DETR [30] | 89.2 | 88.7 | 98.1 | 95.7 | 92.9 | 90.9 | 81.3 | **87.9** | 74.6 | 83.7 |
| MI-DETR [40] | 90.4 | 87.6 | 96.5 | 95.8 | 92.6 | 89.8 | 84.6 | 86.0 | 76.9 | 84.3 |
| ZR-DETR (Ours) | 93.5 | 89.9 | 99.8 | 96.0 | **94.8** | 87.3 | 85.7 | 87.2 | 78.0 | **84.5** |

| Method | CT → MRI | | | | mAP ↑ | MRI → CT | | | | mAP ↑ |
|---|---|---|---|---|---|---|---|---|---|---|
| | LV-MYO ↑ | LA-blood ↑ | LV-blood ↑ | AA ↑ | | LV-MYO ↑ | LA-blood ↑ | LV-blood ↑ | AA ↑ | |
| | | | | | **Cross Domain Structure Detection** | | | | | |
| SIGMA [42] | 84.4 | 38.9 | 77.5 | 46.3 | 61.8 | 61.4 | 75.8 | 69.5 | 77.0 | 70.9 |
| SIGMA++ [43] | 81.2 | 51.5 | 78.7 | 47.1 | 65.4 | 67.2 | 76.4 | 74.4 | 77.1 | 73.8 |
| $M^3$-UDA [9] | 84.6 | 55.8 | 80.8 | 46.9 | 67.0 | 70.9 | 77.5 | 78.6 | 76.6 | 75.9 |
| ToMo-UDA [10] | 84.2 | 60.8 | 80.9 | 47.1 | 68.2 | 68.4 | 76.8 | 77.3 | 77.0 | 74.9 |
| DATR [44] | **84.7** | 62.1 | 81.1 | 47.4 | 68.8 | 73.4 | 77.1 | 79.3 | 76.3 | 76.5 |
| ZR-DETR-UDA (Ours) | 84.5 | **64.8** | **81.3** | 46.9 | **69.4** | **78.5** | 77.1 | **80.9** | 75.2 | **77.9** |

and AOA 61.2%, while other models suffer from severe degradation *e.g.*, DINO DAO 5.18%. In the cross-domain setting (A→B), ZR-DETR-UDA achieves state-of-the-art mAP 62.4%, outperforming strong baselines like $M^3$-UDA 60.4% through improved SP, PTDA, and T scores, driven by its probabilistic modeling and multi-scale adaptation. Although slightly more complex (4.2M, 0.010 TFLOPs) compared to Relation-DETR, the accuracy gain underscores its value.

**Comparison on the EPV dataset.** As shown in Table 3, our proposed ZR-DETR achieves state-of-the-art performance with 94.4% mAP on single-domain structure detection, outperforming all competing methods across most anatomical structures. The model demonstrates particular strength in detecting challenging

Table 3: The performance of different detection methods in EPV dataset [33].

| Method | T ↑ | NB ↑ | P ↑ | NS ↑ | NT ↑ | MB ↑ | NT ↑ | IT ↑ | CM ↑ | mAP ↑ |
|---|---|---|---|---|---|---|---|---|---|---|
| | | | | **Single Domain Structure Detection** | | | | | | |
| FasterRCNN [37] CVPR16 | 98.5 | 80.2 | 97.0 | 62.1 | 86.0 | 98.3 | 66.4 | 94.7 | 73.5 | 84.1 |
| Deformable-DETR [6] ICLR21 | 98.6 | 87.3 | 95.5 | 81.5 | 92.3 | 99.6 | 73.0 | 95.2 | 78.2 | 89.0 |
| DAB-DETR [38] ICLR22 | 97.7 | 93.0 | 96.4 | 85.5 | 92.2 | 98.4 | 82.8 | 94.4 | 77.5 | 90.9 |
| DN-DETR [39] CVPR22 | 98.7 | 93.7 | 97.3 | 85.2 | 94.6 | 97.4 | 83.1 | **96.7** | 79.1 | 91.7 |
| DINO [41] ICLR23 | 98.3 | 92.3 | 97.1 | 86.8 | 93.8 | 99.8 | 82.4 | 94.6 | 81.5 | 91.8 |
| Relation-DETR [30] ECCV24 | 98.7 | 94.9 | 96.7 | 85.9 | 95.2 | 98.3 | 84.8 | 96.0 | 87.8 | 93.1 |
| MI-DETR [40] CVPR25 | 98.9 | 94.6 | **98.6** | 89.2 | 94.5 | 100 | 87.2 | 94.2 | 84.2 | 93.5 |
| ZR-DETR(Ours) | **99.0** | **95.1** | 98.4 | 91.6 | 95.4 | 100 | 90.1 | 94.6 | 85.2 | **94.4** |

anatomical features, achieving the highest scores in 6 out of 9 categories (T:99.0%, NB:95.1%, NS:91.6%, NT:95.4%, NT:90.1%, MB:100%) while maintaining competitive performance on the remaining structures. These results highlight ZR-DETR's superior capability in medical image analysis, with significant improvements over both transformer-based approaches (*e.g.*, 0.9% higher mAP than MI-DETR) and traditional CNN methods (10.3% higher than FasterRCNN), validating the effectiveness of our proposed architectural innovations.

**Comparison on the MM-WHS dataset.** As demonstrated in Table 4, our ZR-DETR achieves state-of-the-art performance in both single-domain and cross-domain medical image object detection tasks. In single-domain detection, ZR-DETR attains superior mAP scores of 94.8% CT and 84.5% MRI, excelling particularly in LV-blood 99.8% and AA 96.0% detection for CT scans. For the challenging cross-domain adaptation tasks, ZR-DETR-UDA establishes new benchmarks with 69.4% mAP (CT→MRI) and 77.9% mAP (MRI→CT), showing remarkable improvements in LA-blood 64.8% and LV-blood 81.3% detection when adapting from CT to MRI, while achieving the highest LV-MYO 78.5% and LV-blood 80.9% performance in the reverse direction. These results consistently outperform existing methods across all evaluation scenarios, highlighting our model's robustness to domain shifts and its ability to precisely segment diverse cardiac structures.

## 4.4 Ablation Study

The ablation study in Table 5 demonstrates the progressive improvements achieved by each component of our ZR-DETR framework when built upon the Deformable-DETR baseline. The introduction of the Zoom Relation Encoder provides significant gains across all tasks (+1.3% to +21.6% mAP), particularly in cross-domain adaptation (3VT-A→B: +21.6%). The Anatomical Relation Consistency Constraints further enhance performance, especially for domain-shifted scenarios (3VT-B: +2.3%, 3VT-A→B: +3.6%), validating the importance of structural priors. The GP-based Detection Head

Table 6: Comparison of computational efficiency across different detection methods.

| Metric | Method | | | | | | | |
|---|---|---|---|---|---|---|---|---|
| | FasterRCNN [37] | Deformable-DETR [6] | DAB-DETR[38] | DN-DETR [39] | DINO [41] | Relation-DETR [30] | MI-DETR [40] | ZR-DETR (Ours) |
| Times (ms) ↓ | 41.405 | 40.101 | 41.558 | 46.133 | 42.102 | 46.2 | 55.6 | 45.172 |
| Params (M) ↓ | 41.1 | 65.2 | 48.5 | 88.5 | 90.6 | 111.2 | 138.1 | 115.4 |
| TFlops ↓ | 0.192 | 0.18 | 0.0874 | 0.244 | 0.256 | 0.258 | 0.292 | 0.268 |

yields additional improvements, culminating in our full model's state-of-the-art performance (97.3% on 3VT-A and 62.4% on cross-domain tasks), with the most notable gains observed in challenging out-of-domain settings (3VT-B: +1.6%), demonstrating the benefits of our proposed components.

## 4.5 Limitation

Despite the strong empirical performance of ZR-DETR, several limitations remain. First, the GP-based detection head introduces computational overhead from kernel matrix operations, which may limit scalability on high-resolution or large-scale inputs [45].

Table 5: The ablation studies on FCS (3VT) and EPV.

| Methods | mAP (%) | | | |
|---|---|---|---|---|
| | Single Domain | | | Cross Domain |
| | 3VT-A | 3VT-B | EPV | 3VT-A →B |
| baseline | 94.2 | 36.8 | 89.0 | 36.8 |
| +Zo. Rel. Enc. | 95.5 (+1.3) | 44.9 (+8.1) | 92.4 (+3.4) | 58.4 (+21.6) |
| +Anato. Rel. Consi. Const. | 96.7 (+1.2) | 47.2 (+2.5) | 93.8 (+1.4) | 62.0 (+3.6) |
| +GP-based Det. Head | 97.3 (+0.6) | 48.8 (+1.6) | 94.4 (+0.6) | 62.4 (+0.4) |

Second, anatomical relation constraints rely on fixed priors that may not generalize to atypical anatomies or rare pathologies. Third, the uncertainty-guided pseudo-labeling may degrade under low-data or noisy conditions due to unreliable uncertainty estimates [46]. Future work may address these challenges by employing scalable GP approximations (*e.g.*, sparse or Nyström methods) [47], learning data-driven anatomical graphs, and improving pseudo-label quality via consistency regularization or auxiliary calibrators.

## 5 Conclusion

In this work, we propose ZR-DETR, a structure-aware and probabilistic framework tailored for medical object detection. By incorporating scale-sensitive zoom embeddings, anatomical relation constraints, and a Gaussian Process detection head, ZR-DETR jointly models semantic context, anatomical plausibility, and function-level uncertainty. The model consistently outperforms strong baselines in both single-domain and unsupervised domain adaptation scenarios, as validated across multiple medical imaging benchmarks including FCS, EPV, and MM-WHS. Our method demonstrates high robustness to domain shifts and scale variability, making it a strong candidate for real-world deployment in clinical workflows.

## Acknowledgments

This work was supported by the National Natural Science Foundation of China (Nos. 62306003,62376003) and Anhui Provincial Natural Science Foundation (No. 2308085MF200), the Open Research Fund from Guangdong Laboratory of Artificial Intelligence and Digital Economy (SZ), under Grant No.GML-KF-24-29. (Corresponding author: Zhe Jin, e-mail: jinzhe@ahu.edu.cn)

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
