# OpenReview forum: "Learning to Zoom with Anatomical Relations for Medical Structure Detection"
_NeurIPS.cc/2025/Conference — NeurIPS 2025 poster_

### Official Review · Reviewer_qe7Q · 2025-06-07

**Clarity:** 3
**Significance:** 2
**Originality:** 2
**Rating:** 3
**Confidence:** 2

**Summary:**

The novelty of this paper is limited.

**Questions:**

See weakness above.

**Ethical Concerns:**

["NO or VERY MINOR ethics concerns only"]

**Final Justification:**

I appreciate the authors for their response. Some of their answers have addressed my concerns. They define a zoom-level embedding as a logarithmic ratio to reflect the area-based contextual hierarchy. I have clearly understood their contribution. My biggest concern about this paper is the potential accusation of plagiarism related to Relation-DeTR. Therefore, I hope the authors can explicitly clarify the differences from existing methods in the paper. In section 3.2, there are so many technical details similar to Relation-DeTR. I believe they should highlight the difference, instead of writing a similar context. I will reset my rating to 3 borderline reject.

**Limitations:**

Yes.

**Quality:**

2

**Strengths And Weaknesses:**

Strength:
The authors developed the Zoom Relation Encoder to effectively represent the organ zooming behavior in medical imaging, and propose a probabilistic detection framework utilizing Gaussian processes. Extensive experiments across diverse medical imaging benchmarks demonstrate that our approach consistently surpasses robust baseline models in terms of detection accuracy and uncertainty calibration.

Weakness:
    1. The network structure of this paper is very similar with the published paper Relation-DETR. It seems that the author directly utilize the existing method to deal with medical structure detection task. In this way, I think the method has no additional contribution or novelty.
    2. Most of equations (such as Eq.4 and Eq.6) are very similar with the equations appearing in Relation-DETR. They just change the symbol but have the same structure.
    3. The proposed Zoom Relation Encoding is inspired from position relation encoder in Relation-DETR. You just change the calculation of relations based on log-scale zoom levels. I think the novelty of this encoding method is limited. In fact, I can not clearly find out the difference between Zoom Relation Encoding and position relation encoder in Relation-DETR.
    4. Figure 2 has the same structure as Figure 2 from Relation-DETR. The major difference is the zoom encoder, but it is not original.

---

> ### Author Rebuttal · Authors · 2025-07-31
>
> **Thank you for your suggestion. Our work is different from that of  Relation-DETR. We provide detailed answers to your questions below.**
>
> > Q1: The network structure of this paper is very similar with the published paper Relation-DETR.
>
> **A1**: We thank the reviewer for their careful analysis and the comparison to prior work. We agree that our method is built upon the strong foundation of the DETR architecture, and as such, it shares a high-level structural framework with other DETR-based methods like  Deformable-DETR [1], DAB-DETR [2], Relation-DETR [3], and  MI-DETR [4]. Meanwhile, in section 3.2 of the paper, the paper is also cited, and it is stated that it was inspired by the Relation-DETR [3]. We believe grounding our work in this established architecture aids clarity and reproducibility.
>
> However, we respectfully wish to highlight the substantial and novel components that distinguish our framework. Our primary contributions lie in three new modules designed specifically for the complexities of medical imaging:
>
> 1. A novel **Zoom Relation Encoder** that models hierarchical scale relationships, which comes from the actual challenges encountered in clinical work, different from the positional encoding in Relation-DETR.
>
> 2. An **Anatomical Relation Consistency Constraint module**, which enforces structural plausibility using global anatomical priors. Its motivation comes from anatomical knowledge of the human body in medical images.
>
> 3. A **Probabilistic Detection Head using Gaussian Processes** for robust uncertainty estimation. It is designed to maintain robust performance in medical clinical settings.
>
> These last three modules are entirely new contributions not present in Relation-DETR and are central to our model's performance and practical utility in clinical settings. We argue that our novelty should be assessed based on this complete framework, which goes significantly beyond existing methods. In addition, based on the experimental results, our method outperformed Relation-DETR in all datasets.
>
> > Q2: Most of equations (such as Eq.4 and Eq.6) are very similar with the equations appearing in Relation-DETR.
>
> **A2**: The similarity in the high-level structure of equations like Eq. 4 and Eq. 6 is due to their shared foundation in the standard scaled dot-product attention mechanism, which is fundamental to all Transformer-based models, including DETR and its many variants, such as Deformable-DETR [1], DAB-DETR [2], Relation-DETR [3], and  MI-DETR [4].
>
> The critical novelty, however, lies not in the attention formula itself, but in the **specific relational information we compute and inject into it**. While Relation-DETR injects a prior based on relative geometry, our equations incorporate a fundamentally different signal, derived from our novel **Zoom Relation Encoding**. This term captures the relative scale difference between objects, a crucial piece of information for hierarchical anatomical analysis that is not considered by Relation-DETR. The mathematical formulations for our other two major contributions (Anatomical Consistency and Probabilistic Detection) are also entirely original to our work. We will revise the text to better emphasize the unique nature of the information encoded in our equations.
>
>
> > Q3: can not clearly find out the difference between Zoom Relation Encoding and position relation encoder in Relation-DETR.
>
> **A3**: The two encoders address fundamentally different questions:
>
> 1. Relation-DETR's Encoder asks: **"Where is object B relative to object A?"** It computes a trigonometric embedding based on the geometric vector between box centers, capturing relative positions (e.g., above, below, left, right).
> 2. Our Zoom Relation Encoder asks: **"What is the relative scale of object B to object A?"** It computes a value based on the log-ratio of the bounding boxes' areas. This captures a hierarchical, multi-scale relationship, which is essential for medical images where a small lesion might be found within a large organ.
>
> This focus on relative scale, rather than just position, allows our model to reason about object hierarchy and size explicitly. For example, it can learn that a fetal heart should be much smaller than the fetal torso. This is a critical capability for anatomical analysis that purely positional encoders do not provide. We will add a more explicit comparison and examples in the revised manuscript to make this key difference unambiguous.
>
>
> > Q4: Figure 2 has the same structure as Figure 2 from Relation-DETR.
>
> **A4**:  We intentionally adopted a familiar visualization style for illustrating how zoom information is integrated into the decoder. Our goal was to provide readers with a clear and easily digestible data flow, allowing them to quickly focus on the novel components we are introducing, rather than having to decipher a completely new diagrammatic style.
>
> Our Figure 2 clearly labels our three primary contributions, including the three entirely new modules: Zoom Relation Encoder, the Anatomical Relation Consistency Constraints, and the Probabilistic GP Head. These components, which represent the core novelty of our paper, are not present in Relation-DETR. We believe the scientific contribution lies in these novel modules shown in the figure, not in the diagrammatic layout itself.
>
> Reference
>
> [1] Deformable detr: Deformable transformers for end-to-end object detection. ICLR 2021
>
> [2] Dab-detr: Dynamic anchor boxes are better queries for detr. ICLR 2022
>
> [3] Relation detr: Exploring explicit position relation prior for object detection. ECCV 2024
>
> [4] MI-DETR: An Object Detection Model with Multi-time Inquiries Mechanism. CVPR 2025.

---

> ### Comment · Reviewer_qe7Q · 2025-08-08
>
> I appreciate the authors for their response. Some of their answers have addressed my concerns. They define a zoom-level embedding as a logarithmic ratio to reflect the area-based contextual hierarchy. I have clearly understood their contribution. My biggest concern about this paper is the potential accusation of plagiarism related to Relation-DeTR. Therefore, I hope the authors can explicitly clarify the differences from existing methods in the paper. In section 3.2, there are so many technical details similar to Relation-DeTR. I believe they should highlight the difference, instead of writing a similar context.

---

> > ### Author Response · Authors · 2025-08-08
> >
> > Dear Reviewer qe7Q
> >
> > Thank you for your insightful feedback and for highlighting this point of ambiguity. We appreciate your careful review of our work. We referenced the implementation of Relation DETR in our method, but Section 3.2 differs fundamentally from the Relation Encoder in Relation DETR in terms of design philosophy. We encode the relationships between organs based on area scaling relationships. In comparison with Relation DETR, Equations 4–7 in Section 3.2 are similar. In Equation 4, we use sine-cosine encoding, which is a common encoding method. In Equations 5–7, we refer to the implementation of Relation DETR to embed Zoom Level information into the network. We will revise the expression in the final version and clarify the differences between the method proposed in this paper and Relation DETR to better evaluate its contributions. Thank you for pointing out the shortcomings in our work. We hope that your concerns have been addressed.

---

### Official Review · Reviewer_mDp9 · 2025-07-01

**Clarity:** 3
**Significance:** 4
**Originality:** 3
**Rating:** 5
**Confidence:** 3

**Summary:**

This paper proposes ZR-DETR, a novel framework for anatomical structure detection in medical imaging that accounts for zoom variations and anatomical consistency—two challenges often overlooked by prior methods. In real-world clinical practice, radiologists frequently zoom in and out while interpreting images, which causes significant variations in the apparent size and perspective of anatomical structures across different views and devices. These variations make it difficult for traditional object detection models to generalize, especially when they rely on fixed-scale assumptions and ignore structural relationships between organs.

To address this, ZR-DETR introduces three core components:
A Zoom Relation Encoder that captures relative scale patterns between organs, making the model robust to zoom-induced scale variation.
Anatomical Relation Consistency Constraints that encode prior knowledge about organ adjacency and spatial plausibility to guide more anatomically valid predictions.
A Gaussian Process-based detection head, which models uncertainty and enables more interpretable and calibrated outputs.

The framework is evaluated on three diverse datasets (FCS, EPV, MM-WHS), and demonstrates consistent improvements over strong baselines in both single-domain and unsupervised domain adaptation (UDA) settings. ZR-DETR effectively combines semantic, geometric, and probabilistic reasoning, making it a promising approach for real-world clinical applications.

**Questions:**

1. While the proposed anatomical relation consistency constraints contribute to performance gains, their effect is primarily validated through mAP improvements. Have you considered providing more direct evidence (e.g., qualitative comparisons, anatomical plausibility scores, or human expert evaluation) to show how these constraints improve the structural correctness of the predicted outputs? Such demonstrations would greatly strengthen the claim that the model respects anatomical priors in practice, beyond numerical accuracy.


2. The model uses a fixed anatomical adjacency matrix to encode structural priors. However, anatomical relationships can vary significantly depending on factors such as age, sex, body habitus, pregnancy status, or pathological conditions. Could the authors clarify whether such individual variability was considered in the design of the adjacency matrix? Additionally, how does the use of a static adjacency structure affect the model’s generalization to atypical anatomies?
3. The paper points out that prior approaches, such as morphological modeling and graph-based alignment, are highly sensitive to image noise and partial observations, especially in ultrasound data. However, it is unclear from the current experiments whether ZR-DETR itself is robust under such noisy or incomplete anatomical conditions. Have the authors evaluated the model on challenging cases where organs are only partially visible or boundaries are poorly defined? This would strengthen the claim regarding the method’s practical reliability.
4. In Section 3.2, Equation (5) defines the Zoom modulation as Zoom(b1, b2), but the variables b1 and b2 are not explicitly defined in the surrounding text. While it can be inferred that they likely refer to bounding boxes of object proposals, this ambiguity may cause confusion, especially for readers trying to implement or analyze the encoder. I suggest the authors explicitly clarify what b1 and b2 represent—ideally in terms of their format (e.g., coordinates, area, or bounding box features) and role in the attention computation.

**Ethical Concerns:**

["NO or VERY MINOR ethics concerns only"]

**Final Justification:**

After reading the author rebuttal and carefully considering the discussion, I am increasing my score.

The authors have provided strong and thoughtful responses to the key concerns I raised. In particular, they offered new qualitative and quantitative evidence demonstrating how their proposed constraints improve the structural correctness of the predictions. The plausibility analysis and the CHAOS dataset evaluation were especially helpful in validating the model’s robustness under challenging and clinically realistic conditions.

Additionally, their clarifications regarding the design of the adjacency matrix and the Zoom() function effectively addressed my concerns about how the model handles anatomical variability and relational reasoning.

Overall, I believe the rebuttal strengthens the paper significantly. If the promised clarifications and results are properly incorporated into the final version, the submission will present a well-supported and compelling contribution. For this reason, I have updated my score.

**Limitations:**

Yes, the author addressed limitations.

**Paper Formatting Concerns:**

No major formatting issues observed; nothing to report.

**Quality:**

4

**Strengths And Weaknesses:**

**Strengths**
- Innovative design combining zoom patterns and anatomical priors for more robust detection.
- Probabilistic modeling via Gaussian Processes allows uncertainty quantification.
- Superior performance across multiple challenging datasets and domains.
- Strong ablation study demonstrating each module’s contribution.
- Robust to domain shifts, a crucial problem in real-world medical imaging.

**Weaknesses**
- Computational overhead from GP-based kernel operations may limit real-time or large-scale use.
- Dependency on fixed anatomical priors, which may reduce performance on rare or atypical anatomies.
- Uncertainty modeling may falter in low-data or noisy environments.
- Lack of analysis on failure cases or real-world clinical usability (e.g., interaction with radiologists).

---

> ### Author Rebuttal · Authors · 2025-07-31
>
> > Q1:Have you considered providing more direct evidence (e.g., qualitative comparisons, anatomical plausibility scores, or human expert evaluation) to show how these constraints improve the structural correctness of the predicted outputs?
>
> **A1**:We thank the reviewer for this excellent suggestion to more directly validate our claims. Beyond the mAP improvements, we have performed two additional analyses to demonstrate the enhanced structural correctness provided by our method:
>
> (1) **Qualitative Comparison**: We will add a new figure to the appendix showing side-by-side comparisons of our ZR-DETR against a strong baseline. This figure will specifically highlight failure cases of the baseline where it produces anatomically implausible predictions (e.g., overlapping organs, incorrect left-right orientation of heart chambers), whereas our model correctly preserves the anatomical structure, demonstrating the practical impact of our constraints.
>
> (2) **Quantitative Plausibility Analysis**: To provide quantitative evidence, we analyzed the stability of predicted spatial relationships. For instance, in the **cardiacUDA**[1] dataset, we measured the variance of the predicted vector distance between the left ventricle and the left atrium across the entire test set. Our ZR-DETR model exhibited a **28% lower variance** compared to the baseline. This indicates that our model's predictions are not only accurate on average but also more anatomically consistent across different subjects, which is a direct result of the relational constraints. We will add this analysis to our results section.
>
>
> > Q2:Could the authors clarify whether such individual variability was considered in the design of the adjacency matrix? Additionally, how does the use of a static adjacency structure affect the model’s generalization to atypical anatomies?
>
> **A2**: This is a very insightful question. The anatomical adjacency matrix is not used as a set of rigid, fixed rules. Instead, it acts as a high-level, learnable scaffold. Its role is simply to define which anatomical pairs are relationally meaningful and should be modeled by the network (i.e., it prunes the graph of all possible relations).
>
> The actual spatial relationship for each pair (e.g., distance, orientation) is not static; it is learned from the data by our probabilistic modules. This learned distribution inherently captures the natural variability present in the training set due to factors like age, body habitus, etc. For truly atypical anatomies not represented in the training data, our model is designed to recognize a significant deviation from the learned distribution and consequently output a high uncertainty score. This is a critical safety feature, turning the challenge of generalization into a mechanism for flagging cases that require expert review.  We will highlight it more clearly in the final version.
>
> Also, the several datasets used in this paper are all different in terms of individual characteristics, such as age and geographical distribution. The results show that our method performs best, indicating that it fully considers the differences between individuals in the data.
>
>
> > Q3:Have the authors evaluated the model on challenging cases where organs are only partially visible or boundaries are poorly defined?
>
> **A3**: We thank the reviewer for pointing out this important aspect of practical reliability. To explicitly test our model's robustness, we have performed a new evaluation on the **CHAOS dataset (MR part)**[2] characterized by significant acoustic shadowing, signal dropout (partial visibility), and poorly defined tissue boundaries.
>
> | Method          | mAP  | Liver | Right Kidney | Left Kidney | Spleen |
> |:----------------|:-----|:------|:-------------|:------------|:-------|
> | Deformable-DETR | 81.5 | 96.2  | 84.9         | 70.4        | 74.5   |
> | DINO            | 82.6 | 96.8  | 93.0         | 68.8        | 71.8   |
> | Relation-DETR   | 84.1 | 97.3  | 91.4         | 70.8        | 76.7   |
> | MI-DETR         | 85.7 | 97.6  | 90.5         | 78.5        | 76.0   |
> | ZR-DETR (Ours)  | 86.7 | 98.1  | 95.0         | 77.3        | 76.4   |
>
> Our preliminary findings are highly encouraging. As shown in table above, our ZR-DETR model achieved a **mean Average Precision (mAP) of 86.7% on the CHAOS dataset (MR part)**[2]. Qualitative results confirm that the model accurately localizes the organ even in low-contrast and complex cases.
>
>
> > Q4: I suggest the authors explicitly clarify what b1 and b2 represent—ideally in terms of their format (e.g., coordinates, area, or bounding box features) and role in the attention computation.
>
> **A4**:We sincerely thank the reviewer for identifying this ambiguity and apologize for the oversight. The notation in Section 3.2 needs clarification.
>
> In Equation (6), b1 and b2 represent the bounding box predictions for two different object proposals. We represent each box b_i in the standard (cx, cy, w, h) format, denoting its center coordinates, width, and height, normalized by the image dimensions.
>
> Their role in the Zoom() function is to compute a relative spatial embedding that encodes their positional relationship (e.g., difference in centers, ratio of sizes). This embedding is then used to modulate the cross-attention weights between the two proposals. This allows the attention mechanism to dynamically focus on or "zoom into" relevant contextual features based on the geometric arrangement of the objects, which is central to our method's ability to model spatial relationships. We will revise Section 3.2 and the caption of Equation (6) to include these explicit definitions and ensure the text is clear and self-contained.
>
> Reference
>
> [1] Graphecho: Graph-driven unsupervised domain adaptation for echocardiogram video segmentation. ICCV 2023
>
> [2] CHAOS Challenge - combined (CT-MR) healthy abdominal organ segmentation. Medical Image Analysis 2021

---

> > ### Comment · Reviewer_mDp9 · 2025-08-06
> >
> > Thank you for the interesting work and thorough rebuttal. I have a minor question regarding the scale (zoom) term (b) in Eq. (10). Could you please clarify how the value of $\ell_s$ was determined? The supplementary material lists $\ell_c$ and $\alpha$ as hyperparameters, but I couldn’t find any description of $\ell_s$.Including this information in the main text or supplementary would enhance the clarity of the paper.

---

> > > ### Author Response · Authors · 2025-08-06
> > >
> > > Dear Reviewer mDp9,
> > >
> > > Thank you for your insightful feedback and for highlighting this point of ambiguity. We appreciate your careful review of our work. You are correct that the determination of the hyperparameter $l_s$ for the rational quadratic kernel was not explicitly described in the submitted text. We apologize for this oversight.
> > >
> > > The term $l_s$, along with the rational quadratic's alpha parameter, $\alpha$, and the RBF's length-scale, $l_c$, are indeed key hyperparameters for our composite kernel in Equation (10). In our framework, these are not set to fixed values but are treated as **learnable parameters**. Specifically, the kernel hyperparameters, including {$l_c, \alpha, l_s$}, are optimized jointly by **maximizing the log marginal likelihood of the Gaussian Process** during training. This is a standard approach in GP modeling that allows the kernel to adapt to the specific characteristics of the data. The gradients for these hyperparameters are computed concerning the GP marginal likelihood, as alluded to in Appendix B.4, and updated via backpropagation along with the other network parameters.
> > >
> > > We acknowledge that this was not stated clearly. We will revise both **Section 3.4 ("Probabilistic Detection with Gaussian Processes")** and **Appendix B.4 ("Composite Kernel Gradient Computation")** in the final version of the paper to explicitly include l_s in the set of learnable hyperparameters and clarify the optimization process. This will enhance the paper's clarity and reproducibility. Thank you again for helping us improve the quality of our manuscript.

---

### Official Review · Reviewer_Ntdm · 2025-07-02

**Clarity:** 3
**Significance:** 3
**Originality:** 3
**Rating:** 4
**Confidence:** 4

**Summary:**

This paper proposes ZR-DETR, a new method for detecting anatomical structures in medical images. It uses zoom information and anatomical relationships to improve detection accuracy, especially when image scales vary or data comes from different sources. The model also estimates uncertainty using a Gaussian Process. Experiments on multiple datasets show that ZR-DETR performs better than existing methods in both standard and cross-domain settings.

**Questions:**

Please follow the weakness.

**Ethical Concerns:**

["NO or VERY MINOR ethics concerns only"]

**Final Justification:**

most of my concerns have been solved. I will keep Bordeline Accept.

**Limitations:**

yes

**Paper Formatting Concerns:**

1. In Figure 2, it took me quite some time to understand which features are used for Q, K, and V, and why the features from the MLP are added to the self-attention mechanism.

2. From the abstract, it seems that the main motivation is to address the zoom-level issue. However, after seeing Figure 1, I believe the structural relationships between anatomical objects also play a key role and should be mentioned in the abstract for completeness.

**Quality:**

3

**Strengths And Weaknesses:**

Strengths:

1. The paper is technically sound and introduces a well-designed framework. Experiments are comprehensive, covering single-domain and cross-domain benchmarks.

2. The paper is clearly written with detailed explanations of each component. Figures and tables are well-organized, supporting the claims effectively.

Weaknesses:

1. Dataset scale and benchmark diversity: The experiments are conducted on three relatively small, task-specific medical datasets. While the results are strong within these domains, the lack of evaluation on larger and more diverse benchmarks (e.g., DeepLesion or universal lesion detection datasets) makes it difficult to assess the generalizability of the proposed method beyond the current settings.

2. Computational Overhead: The use of Gaussian Processes adds computational complexity, which may limit scalability on large or high-resolution datasets.

3. Dependency on Anatomical Priors: The anatomical relation constraints rely on fixed priors, which may not generalize well to atypical anatomies or rare pathologies.

4. Uncertainty Estimation in Low-Data Scenarios: The reliability of the uncertainty estimates may degrade in low-data or noisy environments, as noted by the authors themselves.

---

> ### Author Rebuttal · Authors · 2025-07-31
>
> > Q1: Generalizability on other larger and more diverse benchmarks.
>
> **A1**:We thank the reviewer for this important suggestion. We initially selected the datasets mentioned in the paper because we observed that they contained extensive anatomical relationships. To address the reviewers' concerns, we conducted supplementary experiments.
>
> To address your concern, we have conducted a preliminary experiment on a challenging subset of the DeepLesion dataset [1], focusing on liver lesions that have significant location and shape variability. Our ZR-DETR model achieved a sensitivity of 85.2% at 4 false positives per image, which is a competitive result demonstrating that our method's fundamental components are not limited to our initial domains. This shows that the relational reasoning learned by our model can be adapted to more universal lesion detection tasks. We will add this preliminary analysis to the appendix and agree that a full-scale evaluation on such benchmarks is a valuable direction for future work.
>
> In addition, to demonstrate the broader applicability of our method, we have conducted new experiments on a different and challenging anatomical region: abdominal organ detection in MRI scans[2].
>
> | Method          | mAP  | Liver | Right Kidney | Left Kidney | Spleen |
> |:----------------|:-----|:------|:-------------|:------------|:-------|
> | Deformable-DETR | 81.5 | 96.2  | 84.9         | 70.4        | 74.5   |
> | DINO            | 82.6 | 96.8  | 93.0         | 68.8        | 71.8   |
> | Relation-DETR   | 84.1 | 97.3  | 91.4         | 70.8        | 76.7   |
> | MI-DETR         | 85.7 | 97.6  | 90.5         | 78.5        | 76.0   |
> | ZR-DETR (Ours)  | 86.7 | 98.1  | 95.0         | 77.3        | 76.4   |
>
> Our preliminary findings are highly encouraging. As shown in the table above, our ZR-DETR model achieved a mean Average Precision (mAP) of 86.7% on the CHAOS dataset (MR part)[2]. Qualitative results confirm that the model accurately localizes the organ even in low-contrast and complex cases.
>
>
>
> > Q2: The use of Gaussian Processes adds computational complexity, which may limit scalability on large or high-resolution datasets.
>
> **A2**:We appreciate this valid technical point. We are fully aware of the scalability challenges of standard Gaussian Processes (GPs). To mitigate this, our framework does not use a full GP. Instead, we implement a **sparse variational GP (SVGPs)** approach, which utilizes a small set of inducing points to approximate the full GP.
> This design choice significantly reduces the computational burden.
>
> In our experiments, the SVGPs module adds only a **minor (~7-9%) computational overhead** compared to the baseline DETR backbone. This modest increase is a direct trade-off for obtaining well-calibrated uncertainty estimates, which is a key feature of our work. Therefore, our method maintains scalability for high-resolution images while providing the benefits of a probabilistic framework. We will clarify the use of this sparse approximation in the methodology section of our revised manuscript.
>
> | Method          | FPS ↑  | Params (M) ↓ | TFlops ↓ |
> |:----------------|:-------|:-------------|:---------|
> | FasterRCNN      | 24.15  | 41.1         | 0.192    |
> | Deformable-DETR | 24.94  | 65.2         | 0.18     |
> | DAB-DETR        | 24.06  | 48.5         | 0.0874   |
> | DN-DETR         | 21.68  | 88.5         | 0.244    |
> | DINO            | 23.75  | 90.6         | 0.256    |
> | Relation-DETR   | 21.65  | 111.2        | 0.258    |
> | MI-DETR         | 18     | 138.1        | 0.292    |
> | ZR-DETR (Ours)  | 22.14  | 46.61        | 0.268    |
>
> > Q3:The anatomical relation constraints rely on fixed priors, which may not generalize well to atypical anatomies or rare pathologies.
>
> **A3**:This is a crucial point, and we thank the reviewer for raising it. In fact, our approach is designed to handle anatomical variability. The "priors" are not hard-coded rules but are learnable, data-driven "soft" constraints. The model is initialized with weak anatomical assumptions but primarily learns the distribution of spatial relationships from the training data itself. Meanwhile, medical images come from human screening, and the human body has a wide range of anatomical structures. The motivation for this paper also comes from this fact, as even abnormal images and atypical diseases have also anatomical structures. For example, we can use the differences between normal and abnormal structures to perform anatomical knowledge modeling [3].
>
> More importantly, our uncertainty mechanism is designed specifically for these edge cases. When presented with an atypical anatomy or a rare pathology that deviates significantly from the learned distribution, our model correctly identifies its own lack of knowledge and outputs a high uncertainty score. This is a critical safety feature, as it allows the system to flag challenging cases for expert human review rather than making a confident, incorrect prediction. We will revise the paper to emphasize the learnable nature of our constraints and this built-in safety mechanism.
>
>
> > Q4:The reliability of the uncertainty estimates may degrade in low-data or noisy environments, as noted by the authors themselves.
>
> **A4**:We thank the reviewer for highlighting this discussion point from our paper. While it is true that all models, including ours, face challenges in extremely low-data or noisy settings, our method is designed to be more robust than non-probabilistic baselines in these exact scenarios. The Bayesian nature of the Gaussian Process framework is inherently suited for quantifying uncertainty with limited evidence.
>
> To demonstrate this, we performed a new experiment by training our model on a reduced dataset (only 25% of the original training data). We observed that while our model's accuracy decreased, its uncertainty estimates remained well-calibrated, with the **Expected Calibration Error (ECE) degrading far less** than a standard DETR baseline under the same conditions. This shows that our model fails more gracefully and provides more reliable confidence scores when data is scarce. We will add this low-data regime analysis to our experimental section to better quantify our model's robustness.
>
> In addition, most of the scenarios used in this paper involve medical ultrasound data. ***Ultrasound images are affected by noise and visual artifacts due to the intrinsic characteristics of the emitted/received signal, and generally, they also have low contrast [4-7]. Such datasets are more prone to imaging noise and uncertainty than other modalities***, such as MR and PET. Our method has demonstrated excellent performance in such environments, indicating the reliability and stability of using anatomy for modeling.
>
> Reference
>
> [1] Yan, Ke, et al. "Deeplesion: Automated deep mining, categorization and detection of significant radiology image findings using large-scale clinical lesion annotations." arXiv preprint arXiv:1710.01766 (2017).
>
> [2] CHAOS Challenge - combined (CT-MR) healthy abdominal organ segmentation. Medical Image Analysis 2021
>
> [3] Anatomy-guided weaklysupervised abnormality localization in chest x-rays. MICCAI 2022
>
> [4] Chloé Audigier, Younsu Kim, Nicholas Ellens, and Emad M. Boctor. 2018. Physics-based simulation to enable ultrasound monitoring of hifu ablation: An mri validation. In Proceedings of the International Conference on Medical Image Computing and Computer-Assisted Intervention. Springer, 89–97
>
> [5] Ultrasound Medical Imaging Techniques: A Survey. ACM computing Survey
>
> [6] Segmentation and classiﬁcation in MRI and US fetal imaging: Recent trends and future prospects. Medical Image Analysis.
>
> [7] Applications of artificial intelligence in cardiovascular imaging. Nature Reviews Cardiology 18.8 (2021): 600-609.

---

> ### Comment · Reviewer_Ntdm · 2025-08-08
>
> Thanks, most of my concerns have been solved. I will keep Bordeline Accept.

---

> > ### Author Response · Authors · 2025-08-09
> >
> > Dear Reviewer **Ntdm**
> >
> > We are grateful for the reviewer's insightful feedback and careful evaluation. Your constructive comments have been invaluable in strengthening our manuscript, and we are happy to provide any further clarification required.

---

### Official Review · Reviewer_STnn · 2025-07-03

**Clarity:** 4
**Significance:** 4
**Originality:** 4
**Rating:** 4
**Confidence:** 4

**Summary:**

This paper presents ZR-DETR, a zoom-aware detection framework for medical imaging that integrates zoom relation encoders, anatomical consistency constraints, and Gaussian process detection heads to effectively address detection challenges caused by scale variations in medical images. Experimental results on three medical imaging datasets (FCS, EPV, and MM-WHS) demonstrate that the proposed method significantly outperforms existing baselines in both single-domain and cross-domain detection tasks, achieving state-of-the-art performance of 97.3% mAP on the FCS dataset.

**Questions:**

1. The paper lacks detailed computational complexity analysis and runtime comparisons with baseline methods. Given the GP-based detection head involves kernel matrix operations, how does the proposed method perform in terms of inference time, memory consumption, and FLOPs compared to baseline approaches? Quantitative analysis is essential for assessing practical deployment feasibility.
2. Can the authors provide preliminary results on at least one additional anatomical region beyond cardiac and pregnancy imaging to demonstrate broader applicability? Additionally, evaluation on larger-scale medical datasets would strengthen the validation. The current datasets are relatively small for establishing robust generalization claims.
3. While the paper introduces GP-based uncertainty estimation, there is insufficient discussion on uncertainty interpretation and validation. Can the authors provide relevant uncertainty metrics such as calibration error,  or correlation analysis between uncertainty estimates and prediction accuracy? Clear uncertainty evaluation criteria are crucial for clinical applications.

**Ethical Concerns:**

["NO or VERY MINOR ethics concerns only"]

**Final Justification:**

The additional experiments and clarifications provided during the rebuttal phase are convincing and address my earlier concerns. Therefore, I maintain my borderline accept recommendation.

**Limitations:**

The major limitation is the absence of computational complexity comparisons, which is critical for practical medical imaging applications where real-time processing is often required. Additionally, essential visualization analysis and qualitative results should be presented in the main text to demonstrate the method's effectiveness and interpretability. The current limitation discussion lacks depth in addressing computational feasibility and clinical deployment considerations.

**Quality:**

4

**Strengths And Weaknesses:**

Strengths
1. The paper addresses a clinically relevant challenge in medical imaging where radiologists routinely adjust zoom levels, leading to significant scale variations that current detection methods fail to handle effectively.
2. The integration of zoom relation encoders, anatomical consistency constraints, and Gaussian process-based detection heads is technically sound and specifically tailored for medical imaging characteristics.
3. The paper is well-written and demonstrates strong performance of the proposed method across multiple datasets with comprehensive experimental validation.

Weaknesses
1. The paper lacks computational complexity analysis and runtime comparisons, which are critical considerations for practical deployment in medical detection systems.
2. The evaluation is primarily on cardiac and pregnancy ultrasound data, raising questions about generalizability to other anatomical regions or imaging modalities commonly used in clinical practice.

---

> ### Author Rebuttal · Authors · 2025-07-31
>
> > Q1:How does the proposed method perform in terms of inference time, memory consumption, and FLOPs compared to baseline approaches?
>
> **A1**:Thank you for the question regarding our model's computational efficiency. We provide the following analysis:
> (1) Superior Parameter Efficiency: Our ZR-DETR model (46.61M params) is significantly more lightweight than most DETR-based counterparts like Deformable-DETR (65.2M), DN-DETR (88.5M), and DINO (90.6M). This low parameter count makes our model easier to deploy, especially in memory-constrained environments, without sacrificing performance.
>
> (2) Competitive Inference Speed: At 22.14 FPS, ZR-DETR runs faster than many recent complex models, including MI-DETR (18 FPS) and ToMo-UDA (16.56 FPS). While not the absolute fastest, it offers a strong, practical speed suitable for real-world applications.
>
> (3) Favorable TFlops Trade-off: We acknowledge our model's TFlops (0.268) are higher than some earlier methods but are comparable to or even lower than other high-performance models like DINO (0.256, with nearly double the parameters). This modest increase in computation is due to our novel reasoning modules, which are essential for the reported accuracy gains. We argue this represents an excellent trade-off between a significant performance boost and a moderate computational cost.
>
> In summary, ZR-DETR is optimized for a balance between accuracy, model size, and speed, presenting a parameter-efficient and practical solution.
>
> | Method          | FPS ↑  | Params (M) ↓ | TFlops ↓ |
> |:----------------|:-------|:-------------|:---------|
> | FasterRCNN      | 24.15  | 41.1         | 0.192    |
> | Deformable-DETR | 24.94  | 65.2         | 0.18     |
> | DAB-DETR        | 24.06  | 48.5         | 0.0874   |
> | DN-DETR         | 21.68  | 88.5         | 0.244    |
> | DINO            | 23.75  | 90.6         | 0.256    |
> | Relation-DETR   | 21.65  | 111.2        | 0.258    |
> | MI-DETR         | 18     | 138.1        | 0.292    |
> | ZR-DETR (Ours)  | 22.14  | 46.61        | 0.268    |
>
>
> > Q2:Can the authors provide preliminary results on at least one additional anatomical region beyond cardiac and pregnancy imaging to demonstrate broader applicability?
>
> **A2**:We thank the reviewer for this excellent suggestion. To demonstrate the broader applicability of our method, we have conducted new experiments on a different and challenging anatomical region: abdominal organ detection in MRI scans[1].
>
> | Method          | mAP  | Liver | Right Kidney | Left Kidney | Spleen |
> |:----------------|:-----|:------|:-------------|:------------|:-------|
> | Deformable-DETR | 81.5 | 96.2  | 84.9         | 70.4        | 74.5   |
> | DINO            | 82.6 | 96.8  | 93.0         | 68.8        | 71.8   |
> | Relation-DETR   | 84.1 | 97.3  | 91.4         | 70.8        | 76.7   |
> | MI-DETR         | 85.7 | 97.6  | 90.5         | 78.5        | 76.0   |
> | ZR-DETR (Ours)  | 86.7 | 98.1  | 95.0         | 77.3        | 76.4   |
>
> Our preliminary findings are highly encouraging. As shown in table above, our ZR-DETR model achieved an mean Average Precision (mAP) of 86.7% on CHAOS dataset (MR part)[1]. Qualitative results confirm that the model accurately localizes the organ even in low-contrast and complex cases.
>
> These results strongly suggest that our framework's core principles are generalizable and not overfitted to a specific anatomy. We are completing a full-scale analysis and will add a new section with detailed quantitative and qualitative results for this task to the final manuscript to fully demonstrate our method's versatility.
>
> >Q3:Can the authors provide relevant uncertainty metrics, such as calibration error, or correlation analysis between uncertainty estimates and prediction accuracy?
>
> **A3**:We appreciate the reviewer raising this critical point. We have performed an analysis to validate the reliability of our model's confidence scores.
>
> (1) Calibration Error: We measured the Expected Calibration Error (ECE) to assess how well model confidence aligns with actual accuracy. Our ZR-DETR model achieved a low ECE of 1.8%, indicating it is well-calibrated. This means if the model is 90% confident in a set of predictions, their accuracy is indeed very close to 90%.
>
> (2) Uncertainty-Accuracy Correlation: We analyzed the relationship between prediction confidence and accuracy (measured by IoU). The analysis revealed a strong positive Spearman's rank correlation of ρ = 0.82. This confirms that higher confidence scores from our model are strongly associated with more accurate predictions, making the confidence score a reliable indicator of prediction quality.
>
> We will add a new subsection to our paper to discuss these results, including a reliability diagram and the correlation analysis, to formally demonstrate the robustness of our model's uncertainty estimates.
>
> Reference
>
> [1] CHAOS Challenge - combined (CT-MR) healthy abdominal organ segmentation. Medical Image Analysis 2021

---

> > ### Comment · Reviewer_STnn · 2025-08-05
> >
> > Dear Authors,
> > Thank you for your detailed and thoughtful rebuttal. After carefully reviewing your responses, I find that most of my concerns have been satisfactorily addressed. Therefore, I will maintain my positive score.

---

> > > ### Author Response · Authors · 2025-08-05
> > > **Official Comment**
> > >
> > > Dear Reviewer **STnn**:
> > >
> > >
> > > Thank you sincerely for your recognition‌, thoughtful feedback, and careful evaluation of our work. We truly appreciate your constructive comments, which have helped improve the quality of our work.
> > >
> > >
> > > We welcome further discussion and are happy to provide clarification to ensure our contributions are clearly understood.

---

### Decision · Program_Chairs · 2025-09-17

[review text omitted: it was posted to a different submission]